# Cross-Resistance to Abiraterone and Enzalutamide in Castration Resistance Prostate Cancer Cellular Models Is Mediated by AR Transcriptional Reactivation

**DOI:** 10.3390/cancers13061483

**Published:** 2021-03-23

**Authors:** Iris Simon, Sonia Perales, Laura Casado-Medina, Alba Rodríguez-Martínez, Maria del Carmen Garrido-Navas, Ignacio Puche-Sanz, Juan J. Diaz-Mochon, Clara Alaminos, Pablo Lupiañez, Jose A. Lorente, María J. Serrano, Pedro J. Real

**Affiliations:** 1GENyO, Centre for Genomics and Oncological Research, Pfizer-University of Granada-Andalusian Regional Government, Gene Regulation, Stem Cells & Development Lab, PTS Granada, Avenida de la Ilustracion 114, 18016 Granada, Spain; iris.simon@genyo.es (I.S.); sopero@ugr.es (S.P.); lcasadom@idibell.cat (L.C.-M.); lupi13@correo.ugr.es (P.L.); 2Department of Biochemistry and Molecular Biology I, Faculty of Science, University of Granada, Avenida Fuentenueva s/n, 18071 Granada, Spain; 3GENyO, Centre for Genomics and Oncological Research, Pfizer-University of Granada-Andalusian Regional Government, Liquid Biopsy and Cancer Interception Group, PTS Granada, Avenida de la Ilustracion 114, 18016 Granada, Spain; alba.rodriguez@genyo.es (A.R.-M.); carmen.garrido@genyo.es (M.d.C.G.-N.); jose.lorente@genyo.es (J.A.L.); 4Legal Medicine and Toxicology Department, Faculty of Medicine, University of Granada, Laboratory of Genetic Identification, Avenida de la Investigación 11, 18016 Granada, Spain; 5Universidad Internacional de la Rioja, Avenida de la Paz, 137, 26006 Logroño, Spain; 6Department of Urology, Bio-Health Research Institute (Instituto de Investigación Biosanitaria ibs.GRANADA), Hospital Universitario Virgen de las Nieves, University of Granada, Avenida de las Fuerzas Armadas 2, 18014 Granada, Spain; ignacio.puche.sspa@juntadeandalucia.es; 7GENyO, Centre for Genomics and Oncological Research, Pfizer-University of Granada-Andalusian Regional Government, Nanochembio Lab, PTS Granada, Avenida de la Ilustracion 114, 18016 Granada, Spain; juanjose.diaz@genyo.es; 8Department of Pharmaceutical and Organic Chemistry, Faculty of Pharmacy, Campus de Cartuja, University of Granada, 18071 Granada, Spain; 9Department of Urology, University Hospital of Jaen, Avenida del Ejercito Español 10, 23007 Jaen, Spain; clararamosalaminos@juntadeandalucia.es; 10Comprehensive Oncology Division, Clinical University Hospital, Virgen de las Nieves-IBS, Avenida de las Fuerzas Armadas 2, 18014 Granada, Spain; 11Department of Pathological Anatomy, Faculty of Medicine, University of Granada, Avenida de la Investigación 11, 18016 Granada, Spain; 12Bio-Health Research Institute (Instituto de Investigación Biosanitaria ibs.GRANADA), Personalized Oncology Group, Avenida de las Fuerzas Armadas 2, 18014 Granada, Spain

**Keywords:** castration resistant prostate cancer, androgen receptor, AR-V7, AR-V9, transcriptional regulation, Novel hormonal agents, abiraterone, enzalutamide, cross-resistance

## Abstract

**Simple Summary:**

Prostate cancer is the second most common cancer in males. In prostate cancer cells, androgens bind and activate the intracellular mediator called Androgen Receptor that control cell proliferation and survival. Hormone deprivation therapy is administrated to reduce androgen levels and consequently tumour growth. Unfortunately, most patients develop resistance to hormone treatment over the years and novel hormonal agents, such as Abiraterone or Enzalutamide, are administered. However, many patients do not initially respond or become resistant to these drugs quickly. Firstly, we demonstrated that in hormonal sensitive human prostate cancer cells the combination therapy of Abiraterone plus Enzalutamide reduced cell growth and survival. Moreover, starting from these prostate cancer cell lines, we generated cellular models of resistance to hormonal deprivation alone or in combination with the novel hormonal agents. In all the cases, resistant cell lines restore Androgen Receptor expression, Androgen Receptor functionality, cell proliferation and migration in the absence of androgens. Importantly, these novel cellular models acquire cross-resistance to each other. These results are consistent with clinical trials in castration resistant prostate cancer patients and suggest the biological rationale to test the combination therapy of Abiraterone plus Enzalutamide as first-line treatment in hormone-sensitive prostate cancer patients before becoming hormonal resistant.

**Abstract:**

Androgen deprivation therapy (ADT) and novel hormonal agents (NHAs) (Abiraterone and Enzalutamide) are the goal standard for metastatic prostate cancer (PCa) treatment. Although ADT is initially effective, a subsequent castration resistance status (CRPC) is commonly developed. The expression of androgen receptor (AR) alternative splicing isoforms (*AR-V7* and *AR-V9*) has been associated to CRPC. However, resistance mechanisms to novel NHAs are not yet well understood. Androgen-dependent PCa cell lines were used to generate resistant models to ADT only or in combination with Abiraterone and/or Enzalutamide (concomitant models). Functional and genetic analyses were performed for each resistance model by real-time cell monitoring assays, flow cytometry and RT-qPCR. In androgen-dependent PCa cells, the administration of Abiraterone and/or Enzalutamide as first-line treatment involved a critical inhibition of AR activity associated with a significant cell growth inhibition. Genetic analyses on ADT-resistant PCa cell lines showed that the CRPC phenotype was accompanied by overexpression of *AR* full-length and AR target genes, but not necessarily *AR-V7* and/or *AR-V9* isoforms. These ADT resistant cell lines showed higher proliferation rates, migration and invasion abilities. Importantly, ADT resistance induced cross-resistance to Abiraterone and/or Enzalutamide. Similarly, concomitant models possessed an elevated expression of *AR* full-length and proliferation rates and acquired cross-resistance to its alternative NHA as second-line treatment.

## 1. Introduction

Prostate cancer (PCa) is the most common cancer in non-smoking men worldwide and the third cause of cancer-related death after lung and colorectal cancers (http:/gco.iarc.fr/). Androgen deprivation therapy (ADT) is still the main treatment option for advanced PCa although most patients will eventually develop castration-resistant prostate cancer (CRPC) [1,2].

CRPC patients are frequently treated with novel hormonal agents (NHAs), such as Abiraterone Acetate (AA) and Enzalutamide (Enz) [3,4]. AA blocks testosterone production through 17-α-hydroxylase enzyme (CYP17A1) inhibition [5]. In contrast, Enz binds to the androgen receptor (AR) ligand binding domain (LBD) reducing its nuclear translocation and consequently AR transcriptional activation [6]. However, around 15% of patients are initially unresponsive to both of these treatments and many more acquire resistance 9 to 15 months later [3,4]. Additionally, patients that become resistant to AA develop cross-resistance to Enz and vice versa, challenging the sequential use of these drugs [7,8,9,10,11].

Several molecular mechanisms associated to CRPC and AR have been described: increased testosterone synthesis in the adrenal glands or prostatic tissue, *AR* overexpression, *AR* amplification, *AR* mutations, loss of *AR* expression by hypermethylation of the *AR* promoter or expression of *AR* splice variants (*AR-Vs*) [12,13,14,15,16]. These *AR-Vs* are originated by alternative splicing of cryptic exons located on intron 3 in the *AR* locus, and the resulting protein isoforms conserve the N-terminal activation domain but lose the C-terminal LBD acting as an androgen-independent transcription factor. AR variant 7 (*AR-V7*) is the most commonly studied variant in PCa, and its detection in circulating tumour cells (CTCs) has been described as a prognostic marker for AA and Enz resistance [17]. Recently, Cato et al. showed that AR-V7 forms a heterodimer with AR full-length repressing the expression of relevant tumour-suppressor genes in CRPC cellular models [18]. In addition, *AR-V9* was shown to share a common 3′ terminal cryptic exon with *AR-V7* and was recently described to be co-expressed in AA-resistant PCa metastatic patients [19].

The main aims of this work were to generate and to characterize novel CRPC cellular models from androgen sensitive PCa cell lines: (a) ADT-resistant PCa cell lines (R-ADT) selected in the absence of androgens; (b) Concomitant ADT-NHA-resistant PCa cell lines (R-ADT/AA, R-ADT/E, R-ADT/E + A) obtained through the continuous growth in the presence of NHAs and the absence of androgens. We evaluated the proliferation rates and cell cycle, *AR* expression levels, AR transcriptional activity, functionality (cell migration and invasion) and the cross-resistance among the different NHA therapies in all new CRPC models.

## 2. Material and Methods

### 2.1. Cell Culture

Three different human PCa cell lines were used: LNCaP (androgen-sensitive adenocarcinoma cells derived from supraclavicular lymph node metastasis) and 22RV1 (carcinoma epithelial cell line derived from androgen-dependent CWR22 xenograft after castration-induced regression and relapse), both purchased from the American Type Culture Collection (ATCC, Manassas, VA, USA), and PC-3 (androgen-independent cell line originated from a bone metastasis of prostatic adenocarcinoma), that was kindly provided by Dr Ignacio Gil Bazo (CIMA, Pamplona, Spain) as CRPC model. All cell lines were authenticated using STR at the Laboratory of Genetic Identification (Legal Medicine and Toxicology Department) at the University of Granada. The three cell lines were maintained with an RPMI 1640 medium (BioWest) containing 10% Fetal Bovine Serum (FBS), 1% *MEM Non-Essential Amino Acids Solution* (Gibco), 1% glutaMAX (Gibco) and 1% *Penicillin-Streptomycin Solution 100X* (BioWest) in a humid atmosphere incubator with 5% CO_2_. The cell lines were mycoplasma-free and periodically checked for *Mycoplasma* by the Cell Culture Unit at GENyO.

### 2.2. Generation of Androgen-Deprivation-Treatment-Resistant Cell Lines (R-ADT)

LNCaP and 22RV1 ADT-resistant cell lines (R-ADT) were generated after exposing the parental sensitive cells to a hormone-reduced medium (RPMI + 10% charcoal stripped serum (CSS)) for 6 months (Appendix A). Once the R-ADT cell lines were established, they were treated for five days with: (1) AA (20 μM); (2) Enz (40 μM); and (3) AA + Enz (20 μM + 40 μM, respectively), in order to evaluate the effect of the NHAs as a second-line treatment (Appendix A). The range of concentrations described in the literature for both drugs is very wide: 5–30 μM for AA and 10-80 μM for Enz. We selected an intermediate concentration for each drug considering the physiological concentration administrated to PCa patients.

### 2.3. Generation of Cell Lines Resistant to ADT/NHAs (R-ADT/AA, R-ADT/Enz and R-ADT/AA + Enz) by a Concomitant Use of Treatments

The tumour cells lines resistant to ADT/NHAs were obtained by the continuous exposure of R-ADT cells to increasing AA and/or Enz concentrations. Growth mediums containing fresh NHAs were changed every 5 days in order to maintain a consistent drug concentration during the selection process. To avoid the initial lethality of both treatments, cells were grown in a hormone-reduced medium with increasing treatment concentrations at different time points. Treatment resistance was acquired after 6 months (Appendix A). The final concentrations of the NHAs for resistant cell lines maintenance were: 20 μM for R-ADT/AA; 40 μM for R-ADT/Enz; and 20 μM AA + 40 μM Enz for R-ADT/AA + Enz (Appendix A, respectively).

### 2.4. Treatment with AA or Enz as Second-Line Treatment after Concomitant Therapy (R-ADT/NHAs)

The use of AA or Enz as second-line therapy was done after concomitant treatment (R-ADT/E or R-ADT/AA, respectively). Regarding R-ADT/AA, cells were treated with 40 μM Enz, while for R-ADT/E cells were exposed to 20 μM AA (Appendix A, respectively).

### 2.5. Cell Proliferation Assays

The treatment effect on cell proliferation was evaluated with the real-time cell monitoring assays (RTCA) (xCELLigence; ACEA Biosciences, Inc., San Diego, CA, USA). Cells were monitored on the RTCA system for 5 days, and impedance was recorded as a measurement of Cell Index (CI). At least four experimental replicates for each experimental condition were performed as recommended by the manufacturer.

### 2.6. Cell Cycle Experiments

To study the effect of each treatment in the cell cycle, sensitive and resistant cell lines were dissociated after 5-day cultures, washed with PBS and fixed on ice-cold 70% ethanol. The cells were incubated overnight at –20° C and then incubated with propidium iodide buffer (propidium iodide 50 µg/mL and RNase 100 µg/mL in PBS). The cell cycle distribution was analysed on a BD FACSVerse™. Fluorescent intensity, indicating that N and 2N ploidy were represented as indicators of G_0_/G_1_ and G_2_/M phases, respectively, using the BDFACSuite™, ModFit LT™ and GraphPad Prism™ software.

### 2.7. Cell Migration and Invasion Assays

The cells were resuspended in RPMI + 10% CSS medium at a density of 2 × 10^5^ cells/mL. 200 μL of the cell suspension were seeded on top of a 24-well Transwell with a pore size of 8 μm (Millipore, Bedford, Massachusetts, USA). The lower chamber was filled with 700 μL of RPMI medium supplemented with 10% FBS. The cells were kept under these conditions for 48 h. Then, non-traversed cells from the upper compartment of the transwell were removed using wet swabs. Traversed cells from the lower side of the transwell were fixed in methanol and stained with 0.5% crystal violet solution. For cell invasion assays, the same method was performed, with the exception that a layer of Matrigel^TM^ Matrix (BD Biosciences, NY, USA), simulating an extracellular matrix, was added in the upper chamber. To calculate migration/invasion rates, the total number of cells per insert was determined calculating the number of cells by the area of the microscope-viewing field. An average from 5 random fields at 10× magnification using a microscope (Olympus, Tokyo, Japan) was used to estimate the cell number per field. Then, the total number of traversed cells was recalculated for the entire area of the transwell insert. The results were expressed as the number of traversed cells for every 1 × 10^5^ cells seeded from three independent experimental replicas.

### 2.8. Quantification of AR Full-Length, AR-V7 and AR-V9 Expression and Isoform Sequencing

The total RNA was isolated using TRI Reagent (Life Technologies), and the quality tested in an Agilent Bioanalyzer 2010 (Agilent Technologies). Reverse transcription was performed with 0.5 μg of the total RNA using the Transcriptor First Strand cDNA Synthesis Kit (Roche Life Science). The resulting cDNA was used for qPCR using iTaq Universal SYBR Green Supermix in a HT7900 Fast Real-Time PCR System (Applied Biosystems) using custom primers (Appendix A). Primers were designed according to the structure of the AR isoforms described by Kohli M. et al. [19], and the already described coding sequences (CDS) of the *AR* full-length and *AR-V7.* qPCR conditions were 95 °C for 10 min, followed by 40 cycles of 95 °C for 15 s and 60 °C for 60 s. Standard curves were used to assess the primer efficiency and average change in threshold cycle (ΔCT) values determined for each sample relative to endogenous GAPDH levels and compared to control cultures for fold change calculations 2^(−ΔΔCt)^. The experiments were performed in triplicate to determine the mean standard error, and the student’s *t*-tests were performed with normalization to control for *p*-values.

To confirm the amplification of the desired *AR* splice variants, conventional PCR under the same qPCR conditions was performed, and the PCR products were examined by standard TA subcloning in a pCR2.1 vector (Thermo Fisher Scientific) and Sanger sequenced using the M13 FW primer in an ABI Prism 3130 genetic analyser (Applied Biosystems) (Appendix A).

### 2.9. AR Transcriptional Activity

The transcriptional activity of *AR* was indirectly measured by the qPCR of a selected panel of AR-regulated genes (*CDK1, CDK2, FGF8, FKBP5, KLK3, NDRG1, PMEPA1, TMPRSS2* and *UBE2C*). The qPCR conditions and procedures were as described above, and the designed primers are shown in Appendix A.

### 2.10. Statistical Analysis

All data are expressed as mean ± SD. Statistical comparisons were performed with a paired student’s *t*-test. Values were considered statistically significant at *p* < 0.05.

## 3. Results

### 3.1. Functional and Genetic Analyses of the Response to ADT, AA and Enz as First-Line Therapy in PCa Cells Lines: Wild-Type Models

First, the effect of androgen deprivation therapy (ADT) during 5 days on the three PCa cellular models was analysed. Hormonal-deprivation-produced growth rates decreased in both androgen sensitive cell lines (51% and 24% for LNCaP and 22RV1, respectively), while the PC-3 cell line showed no differences over control cells at the proliferative level (Figure 1A). Regarding the cell cycle analysis, a cell cycle arrest at G_0_/G_1_ for both LNCaP and 22RV1 was observed (Figure 1B and Appendix A). Moreover, the G_0_/G_1_ phase arrest was accompanied by cell death induction detected by the presence of a Sub-G_0_ peak exclusively in LNCaP cells (Figure 1B), indicating that this cell line is more sensitive to ADT than 22RV1. As previously described, no cell cycle arrest or cell death was detected in PC-3 cells, confirming their ADT treatment resistance (Figure 1B and Appendix A).

The genetic analyses showed that, in the case of LNCaP, an increase in the mRNA levels of *AR* total and *AR* full-length was observed, while the expression of most AR target genes was diminished (Figure 1C). For the 22RV1 cell line, the overexpression of *AR* total or *AR* full-length was not observed, while mRNA levels for *AR-V7* and *AR-V9* isoforms were slightly elevated (Figure 1C). In addition, most of the *AR* target genes maintained their expression at mRNA levels (Figure 1C). Finally, in PC-3 cells *AR* total, *AR-V9*, *FKBP5*, *PMEPA1* and *TMPRSS2* levels increased two-fold with respect to the control (wild-type PC-3 grown in FBS) (Figure 1C). Furthermore, *AR-V7* and several AR target genes (*FGF8, KLK3* and *NDRG1*) were not detected.

Once PCa patients develop CRPC, NHA therapies with AA or Enz are administered. To understand the molecular mechanisms behind those therapies, we treated the three PCa cell lines with AA (20 μM), Enz (40 μM) or the combination of both for 5 days. When cell proliferation was analysed, we observed that AA was significantly more efficient reducing proliferation rates than Enz in both LNCaP (20% vs. 41.4%; *p* < 0.05) and 22RV1 cells (41.5 vs. 79.5%; *p* < 0.05). In addition, the combination of Enz with AA (E + AA) had a synergistic effect inhibiting cell growth at a higher rate compared to AA or Enz alone in LNCaP (3.1% vs. 20% for AA, 3,1% vs. 41.4% for Enz; *p* < 0.05) and in 22RV1 (28.9% vs. 41.5% for AA, 28.9% vs. 79.5% for Enz; *p* < 0.05). Again, LNCaP cells are more sensitive to NHA therapy than 22RV1 cells. In PC-3 cells, no differences were observed at the proliferative level, demonstrating that this cell line is also resistant to NHA therapies (Figure 2A).

Furthermore, cell cycle analysis demonstrated that both LNCaP and 22RV1 cell lines are initially sensitive to AA or Enz treatments alone. Both drugs induced G_0_/G_1_ cell-cycle arrest and cell death (SubG_0_ phase), while the combination produced a dramatic cell death increase (Figure 2B). In contrast, PC-3 cell line was resistant to both treatments individually or in combination, as neither G_0_/G_1_ cell-cycle arrest nor cell death was observed in this androgen-independent cell line (Figure 2B).

Concerning the alterations in the AR axis, the expression levels of *AR* total and *AR* full-length in LNCaP under Enz treatment were not repressed compared to control (Figure 2C). However, an important decrease (>50%) in the expression levels of *AR-V7*, *AR-V9* and in the *AR* target genes was found (Figure 2C). Regarding the treatment with AA in the LNCaP cell line, we observed a dramatic decrease of *AR* total, *AR* full-length, *AR-V7* and *AR-V9* isoforms expression, and also, as a consequence, a reduction of the majority of the *AR* target genes studied (Figure 2C). In addition, the comparative effect of AA vs. Enz showed that AA treatment induced higher repression levels in most genes analysed compared with Enz (Figure 2C). Strikingly, the combined E + AA treatment promoted *AR-V7* and *AR-V9* overexpression, while *AR* full-length levels were slightly repressed. Despite the large expression increase of these two isoforms after combined treatment, the expression of most *AR* target genes studied was diminished, suggesting that neither *AR-V7* nor *AR-V9* expression is needed to promote these target genes’ expression changes.

The 22RV1 cell line treated with Enz promoted down-regulation in all genes: *AR* full-length, *AR* total, *AR-V7*, *AR-V9* and *AR* target genes. Conversely, 22RV1 cells treated with AA did not show great variation in the expression of *AR* full-length, *AR* total, *AR-V7* and *AR-V9*, but clearly decreased most of the *AR* target genes. Again, similarly to LNCaP cells, the combined treatment (E + AA) significantly increased the expression of *AR-V7* and *AR-V9* isoforms and, although to a lesser extent, of *AR* full-length and *AR* total (Figure 2C). This was accompanied by a maintained or even increased expression of target AR genes.

Finally, treatments in the resistant cell line PC-3 showed opposite mRNA expression patterns compared to LNCaP and 22RV1. Enz treatment increased *AR-V9*, *FKBP5* and *PMEPA1* expression, whereas *AR-V7* expression disappeared as previously described after ADT treatment (Figure 2C). In contrast, the treatment with AA did not modify the expression of any *AR* isoform, while *AR* target genes expression was induced (Figure 2C). Finally, the combined treatment (E + AA) down-regulated all *AR* isoforms, although *AR* target genes did not show any consistent pattern (Figure 2C).

### 3.2. ADT Resistance Increases AR Transcriptional Activity and Confers Enzalutamide and/or Abiraterone Cross-Resistance (R-ADT Model)

In order to generate a cellular model representing CRPC progression in vitro, LNCaP and 22RV1 cell lines were grown in the absence of steroid hormones (CSS) for 6 months. The generated cell lines, denominated LNCaP R-ADT and 22RV1 R-ADT, were able to grow efficiently in the absence of androgens. LNCaP R-ADT showed a significantly higher proliferation rate than wild-type LNCaP (243.9% vs. 100%, *p* < 0.05), while 22RV1 R-ADT reached a proliferative rate identical to that of their respective wild-type counterparts (103% vs. 100%, n.s.) (Figure 3A). Regarding the cell cycle, both wild-type and R-ADT tumour cell lines showed a similar cell cycle distribution (Figure 3B). Importantly, LNCaP R-ADT cells overcame the ADT-induced cell death from the LNCaP wild-type cell line.

In LNCaP R-ADT, the acquisition of ADT resistance was associated with a six-fold induction of *AR* total and *AR* full length at the mRNA level, while the *AR-V7* and *AR-V9* isoforms were only slightly increased (Figure 3C). Moreover, the mRNA expression of several AR target genes was dramatically increased (*FKBP5*, *NDRG1* and *TMPRSS2*) (Figure 3C). In contrast, the expression of all AR variants (*AR* total, *AR* full length, *AR-V7* and *AR-V9*) increased considerably in 22RV1 R-ADT cells (Figure 3C). Again, this strong induction resulted in a general increase in the expression profile of all AR target genes (Figure 3C).

Next, we wondered whether the acquisition of resistance to ADT conditioned the response to second-generation NHA therapies in PCa cells. For this purpose, AA and Enz were used as second-line treatment after ADT resistance acquisition (Figure 4A). In LNCaP R-ADT cells, the relative growth rate was of 45.8% after AA treatment vs. LNCaP R-ADT alone. Furthermore, a higher tolerance to Enz was acquired in LNCaP R-ADT, showing a relative growth of 55.5% compared with LNCaP R-ADT alone. The combination of Enz and AA (E + AA) was also analysed, and, in this case, we observed a growth rate of 23%. All these results suggest that the acquisition of ADT resistance in the LNCaP cell line promoted a dramatic increase of the tolerance to NHAs as second-line treatments.

Concerning 22RV1 R-ADT, the AA treatment involved a decrease of growth rate to 44.2%, while for the Enz treatment the growth rate remained at 88.5% with respect to control 22RV1 R-ADT (Figure 4B). When the effect of the combined treatment was analysed, proliferation rates were similar to those of the AA treatment alone, suggesting that the effect of Enz was masked by the treatment with AA (39.8% vs. 44.2% for E + AA and AA, respectively) (Figure 4B). Again, ADT resistance increases the survival to NHAs in 22RV1 cells.

### 3.3. Resistance to ADT Combined with NHAs Increases AR Full Length Expression and AR Transcriptional Activity in Both PCa Cell Lines (Concomitant Model: R-ADT/NHAs)

Unfortunately, many CRPC patients treated with Enz or AA develop resistance after 9 to 15 months. We used LNCaP and 22RV1 cell tumour lines to analyse the effect of the concomitant use of ADT in combination with NHAs. After 6 months of selection, cell proliferation and gene expression were evaluated. In the case of LNCaP, when Enz was the NHA used in combination with ADT, the proliferation rate observed in LNCaP R-ADT/E cells was significantly augmented compared with the wild-type cell line used as a control (116% vs. 100%; *p* < 0.05) (Figure 5A). Regarding genetic analyses, we detected a significant increase not only in *AR* total and *AR* full-length expressions but also for *KLK3* and *TMPRSS2* (*p* < 0.05), while most of the remaining AR target genes maintained levels similar to those of wild-type cells (*CDK2*, *FKBBS*, *NDRG1* and *PMEPA1*) (Figure 5B).

In contrast, when AA was concomitant with ADT, we observed that the proliferation rate of LNCaP R-ADT/AA cells significantly decreased in comparison with the LNCaP wild-type cell line (66% vs. 100%, respectively) (*p* < 0.05) (Figure 5A). qPCR analyses showed an increase in the expression patterns of *AR* total and *AR* full-length, while *AR-V7* or *AR-V9* were reduced in LNCaP R-ADT/AA (Figure 5B). Interestingly, LNCaP cells were unable to maintain a stable proliferation under a simultaneous treatment schedule with Enz plus AA concomitant with ADT, since the prolonged exposition to both drugs induced cell cycle arrest and cell death.

In the 22RV1 cell line, the three concomitant resistant models were established. Similarly to LNCaP cellular models, 22RV1 R-ADT/Enz showed higher proliferation rates than 22RV1 wild-type cells (107.9% vs. 100%; *p* < 0.05), while 22RV1 R-ADT/AA cells proliferated significantly less than their control 22RV1 wild-type counterpart (73.1% vs. 100%) (Figure 5A) (*p* < 0.05). Additionally, the effect of the concomitant treatment with double treatment (R-ADT/E + A) showed a significant reduction of the proliferation rate compared with the wild-type cell line (51.9 vs. 100%; *p* ≤ 0.05) (Figure 5A), suggesting that the combination of both NHAs potentiates cell growth inhibition mediated by AA.

Regarding genetic analyses, it is worth highlighting that the concomitant treatment of ADT with AA, Enz or the double treatment on 22RV1 cells showed a general increase of *AR* full-length and *AR-V7* expressions. However, an additional increase of *AR-V9* was observed in 22RV1 R-ADT/AA and 22RV1 R-ADT/E + A, while *AR-V9* expression was not detected in 22RV1 R-ADT/Enz cells (Figure 5B). In general, these three 22RV1 R-ADT cellular models consistently increased the expression of several AR target genes, such as *FKBP5*, *PMEPA1* and *TMPRSS2* (Figure 5B).

### 3.4. Acquisition of ADT Resistance Increases Migration and Invasion in Both PCa Cell Lines

Next, we evaluated whether the acquisition of ADT resistance (R-ADT) and the concomitant exposure to ADT and NHAs affected PCa aggressiveness. Importantly, all three LNCaP cell lines showed a significant increased migration capacity in comparison with the control cell line (LNCaP wild-type) (Figure 6A). This increase was highly prominent in LNCaP R-ADT cells (*p* < 0.001), while it was also observed, to a lesser extent, in both concomitant cellular models: LNCaP R-ADT/E and LNCaP R-ADT/AA cells (*p* < 0.01). In the case of invasion assays, a large increase in invasive capacity was only observed in the LNCaP R-ADT cell line (*p* < 0.01) (Figure 6B). In contrast to migration assays, resistance to ADT plus secondary treatments was not associated to increased invasive abilities (Figure 6B).

Similarly, in 22RV1 cellular models, ADT resistance also produced a large significant increase in migratory capacities (*p* < 0.01) (Figure 6A). All three concomitant cellular models showed a lower increase in cellular migration, being statistically significant only in the presence of Enz alone (22RV1 R-ADT/E) (*p* < 0.01) or in combination with AA (22RV1 R-ADT/E + A) (*p* < 0.05). As previously shown in LNCaP cellular models, only ADT-resistant cells (22RV1 R-ADT) possessed potentiated invasive capabilities (*p* < 0.01) (Figure 6B), while none of the three ADT plus NHA-resistant cell lines showed differences in terms of invasiveness.

### 3.5. Most of the Concomitant PCa Models Developed Cross-Resistance to the Alternative NHA Used as a Second-Line Treatment

Once resistance to concomitant treatment schedules has been achieved, we evaluated the sensitivity of each cell line to the alternative NHA. The proliferation rate in LNCaP R-ADT/E cells treated with AA under ADT conditions was even slightly higher than in the presence of ADT and Enz (117.4% vs. 100%) (Figure 7A left panel), while LNCaP R-ADT/A cells maintained similar proliferation rates in the presence of ADT and Enz or AA treatments (92.6% vs. 100%) (Figure 7A right panel).

Concerning gene expression analysis, the sequential use of AA after the acquisition of resistance to ADT and Enz (LNCaP R-ADT/E + Abiraterone) did not modify the *AR* total or *AR* full-length levels nor most of the *AR* target genes (Figure 7B left panel). However, a down-regulation of the *AR-V7* and *AR-V9* isoforms was detected. On the other hand, when we treated the LNCaP R-ADT/A cells with Enz as a second-line therapy (LNCaP R-ADT/A + Enzalutamide), we observed an increase in *AR* total but not in *AR* full length or the *AR* splicing variants *AR-V7* or *AR-V9*, suggesting that other non-studied alternative *AR* isoforms could be up-regulated. Importantly, those alternative isoforms are not able to increase the gene expression of the evaluated *AR* target genes (Figure 7B right panel).

Similarly to LNCaP, 22RV1 R-ADT/A cells showed an identical proliferation rate when they were grown in the presence of AA or Enz (Figure 7C right panel). However, we observed a significant proliferation rate reduction when we treated the 22RV1 R-ADT/E tumour cell line with AA (R-ADT/E + Abiraterone) (68.7% vs. 100%) (*p* < 0.05) (Figure 7C left panel). From all the four concomitant models evaluated, this is the only one that did not show cross-resistance between Enz and AA treatments.

Finally, qPCR analysis demonstrated that in the case of both 22RV1 concomitant cell lines (22RV1 R-ADT/E and 22RV1 R-ADT/A), the sequential use of NHAs, AA or Enz, respectively, as a second-line treatment promoted a severe down-regulation of all *AR* splicing isoforms and AR target genes (Figure 7D).

In summary, we developed functional and genetic analyses on hormone-sensitive and resistant tumour cell lines, demonstrating that the previous treatment with ADT, and the subsequent resistance acquisition, decreases AA and Enz efficiency. In addition, we also showed that an increased AR transcriptional activity is associated to AA and Enz resistance in the novel PCa cellular models generated in this study (Appendix A).

## 4. Discussion

The idea that androgenic signalling is essential for the growth and maintenance of prostate homeostasis is widely accepted. For decades, ADT has been the most important treatment for men with prostate cancer, especially for those with metastatic disease [1,2]. This is based on the role of the AR and its pathways associated to the promotion of cell growth, proliferation and invasiveness [3]. In clinical practice, androgen deprivation is approached either by bilateral orchiectomy or, more commonly, by the administration of GnRH agonists or antagonists. Nevertheless, the use of ADT in the clinic has led to important controversies. The main one is that ADT does not represent a curative approach. Although it produces responses in up to 95% of men, disease eventually progresses in practically all patients [7]. However, even when the patient develops a castration-resistant status and the disease further progresses, ADT is commonly maintained as a baseline treatment independently of the different sequential lines of treatment. Furthermore, ADT has been associated with significant adverse metabolic effects, sexual dysfunction and/or reduced quality of life [4], making it unclear whether it should be administered continuously or intermittently [6].

There are many evidences that ADT increased *AR* mRNA and AR target genes in CRPC cell lines and primary samples [20,21,22,23,24]. Cai C et al. determined the molecular mechanisms responsible for the transcriptional regulation of AR and its target genes [25]. Similarly, it is reasonable to believe that, in response to NHAs, cells try to compensate for the reduction of androgenic signalling by increasing the expression of AR. However, if the therapy is effective, the transcriptional activity of AR should be reduced. The analysis of the expression levels of a broad panel of well-characterized AR target genes was completed. AR transcriptional repression occurs especially in the case of sensitive cells such as the LNCaP cell line, while in the partially sensitive 22RV1 cell line, the reduction of the transcriptional activity of AR occurs mainly when dealing with second-generation NHAs (Enz and/or AA). It should be noted that LNCaP cells carry an activating AR mutation (T878A) that confers a certain resistance to reduced concentrations of AA (2–5 μM) [26,27]. However, at the working concentration of 20 μM AA used in our study, we observed how, despite having that mutation, LNCaP wild-type cells were highly sensitive to this NHA, as was evident by the G_0_/G_1_ phase arrest and the accumulation of death cells (Sub-G_0_ peak). Moreover, 22RV1 cell lines had a 35kb tandem duplication containing exon 3 from an AR gene [28]. Although 22RV1 cells have been shown to be resistant to 1 μM Enz [29], we observed cell death induction (Sub-G0 peak) after a 5-day 40 μM treatment.

However, the existence of androgen-independent PCa cells such as PC-3 is an example of the fact that the insensitivity to hormonal therapies is a reflection of the variability and complexity of the therapeutic response of prostate cancer patients, depending on the initial characteristics of the tumour cells [30]. While the absence of AR protein in PC-3 androgen-resistant cells is historically well demonstrated [31,32], the sensitivity to Enz has generated more controversy. Our laboratory and many others have shown that PC-3 cells can be considered resistant to this drug [33,34,35,36]. However, very recently, Abadiz A et al. have demonstrated that a high-dose concentration of Enz (30–100 μM) reduces proliferation and induces bax-mediated apoptosis, through the activation of the intrinsic pathway, and the deregulation of the heat shock protein system in PC-3 cells acquired from ATCC [37]. This publication showed for the first time the possible side effects caused by prolonged exposure to Enz in PCa cells lines, regardless of AR expression. In contrast, we have not observed any antiproliferative effect of Enz (40 μM) using RTCA or cell cycle analysis by propidium iodide staining after 5 days of treatment.

Importantly, NHA-mediated cell growth inhibition in hormone-sensitive PCa cell lines was reversed with the acquisition of ADT resistance after 6 months of selective pressure. In fact, we observed that growth rates in the initial androgen-dependent tumour cell lines LNCaP and 22RV1 not only increased but also promoted a more aggressive behaviour, amplifying growth rates even more than the androgen-resistant cell line PC-3.

The implication of AR signalling in the development of ADT resistance in PCa is complex, involving different types of genomic alterations and leading to a host of selective/adaptive responses. For example, ADT induces the reduction of DHT levels, which is the main mediator of AR signalling in prostatic tissue. Consequently, most of the prostate cells undergo apoptosis, but some of them can escape by entering a dormant status [38]. These tumour cells under dormancy are able to adapt to low androgen environment, restarting proliferation with tumour progression [39]. In addition, ADT treatment promotes senescence in tumour cells. In this state cells stop dividing permanently, but are not fully destroyed. Over time, a large amount of senescent cells accumulate in the organs, causing inflammation and damaging neighbouring healthy cells. Senescence was associated with prolonged neo-adjuvant ADT and with relapse and resistance to ADT [40]. These results coincide with our observations, since we detected that in 22RV1 wild-type cells ADT promoted a cell cycle arrest in G_0_/G_1_ that was not accompanied by cell death.

Progression in the CRPC patients involves modifications of genetic pathways directly or indirectly associated to the AR signalling. These modifications include amplification, overexpression, epigenetic changes or mutations of the AR gene. Splicing variants of *AR* have been strongly associated with ADT resistance as well as with AA and Enz resistance [41]. There is a great controversy in the field regarding the correlation of the expression of AR-V7 and the evolution of PCa. Some publications report that an increase in its expression entails lower responses to treatments [2,3,4,5,17], and others suggest that there is no such relationship [6,12,13,14] (For this reason, it is necessary to first understand the role of AR-V7 and other splicing variants contributing to treatment resistance to later standardize the detection methodologies of these isoforms). According to Kohli et al. [19], the cryptic exons CE3 and CE5 are transcribed together, and both appear in the *AR-V9* mRNA. Our experimental design allowed us to detect and differentiate with certainty *AR-V7* and *AR-V9* isoforms. Cloning and sequencing of the two independent amplicons confirmed the effectiveness of our approach and guaranteed the qPCR expression results. We propose that other independent laboratories validate this new strategy in order to standardize AR-Vs detection methodologies and to clarify the current controversy.

In our results, we observed how the tumour cells lines, LNCaP and 22RV1, initially hormone-sensitive, became ADT-resistant after a 6-month treatment. This resistance was accompanied by the overexpression of *AR* full-length but not necessarily by the overexpression of the splice variants, *AR-V7* and *AR-V9*, suggesting that these splice variants might not be essential for the acquisition of ADT resistance. In this context, it was suggested that the growth of tumour cells with high AR-Vs expression did not require the presence of *AR* full-length to induce proliferation of genes associated to AR-Vs [42]. In fact, we detected that in wild-type PCa cells lines, the inhibition of *AR* full-length was associated to an increase of *AR-Vs*. Moreover, *AR-V7* and *AR-V9* isoforms do not always maintain the same pattern of transcriptional regulation with each other. For example, in PC-3 wild-type cells treated for 5 days with Enz, *AR-V7* was totally repressed, while *AR-V9* was slightly induced; on the contrary in 22RV1 R-ADT/E cells both *AR-Vs* followed the opposite regulation pattern. However, all our CRPC cellular models showed AR activation, independently of the AR-V status, in contrast to Cato L et al.’s results in preclinical models, that conclude that AR-V7 heterodimerises with AR full-length and is necessary for CRPC [18,43]. Thus, we consider that it is necessary to analyse all AR variants in order to confirm NHA activities.

The relationship between *AR* full-length and the acquisition of castration resistance was previously evaluated by Shiota M et al. [44]. They identified a high association between the overexpression of *AR* full-length and the Epithelial to Mesenchymal Transition (EMT) process as a new mechanism of castration resistance. Our results demonstrated that the acquisition of ADT resistance increases the ability to migrate, a property acquired during EMT. This characteristic was more evident in LNCaP than in 22RV1 CRPC models. These results coincide with recent results published by Miao L et al. in 2017, who demonstrated that the induction of EMT was an adaptive response to Enz with implications for therapy resistance [45].

Therefore, the question we need to answer is: What is the best treatment combination according to the resistance mechanisms induced by previous treatments [46]? This question is currently the subject of study in several clinical trials. Current findings from COU-301 and AFFIRM trials showed different results for adding AA or Enz as second-line treatment after ADT in metastatic CPRC (mCRPR) patients previously treated with docetaxel. Despite these studies having demonstrated that antiandrogen therapies improve OS and PFS in mCRPC, these patients could eventually progress and develop antiandrogen resistance in the near future. Despite the progress in this field, the question remains open: What is the best NHA to treat mCRPR: AA, Enz, both or none? Our results suggest that the combination of Enz and AA is the most efficient treatment to reduce cell proliferation in both ADT-resistant cellular models, while Enz treatment was the less effective one.

Regarding the potential concomitant use of ADT plus AA or Enz as first-line treatment, the findings from CHAARTED, LATITUDE, and ARCHES trials have demonstrated the improvement of OS and PFS in mCRPC [47]. Our experimental results on the androgen-dependent tumour cell line (LNCaP) showed that the concomitant use of ADT and Enz did not increase treatment efficiency; on the contrary, we detected less ability to reduce cell tumour growth. Similar results were obtained with 22RV1, which exhibited a better tolerance to ADT. However, the combination of AA with ADT showed a significant ability to reduce growth rates, especially in LNCaP, until the emergence of resistance. In addition, once this resistance appeared after the concomitant schedule (R-ADT/A), the sequential use of Enz did not reduce cell proliferation, demonstrating the acquisition of cross-resistance between both NHAs, as previously described in mCRPC patients [7,8,9,10,11]. Moreover, our results are consistent with the low efficacy of AA + Enz combination demonstrated in the Alliance trial in mCRPC patients [48] and represent a biological rationale to test the combination therapy with both NHAs within the hormone-sensitive setting before CPRC acquisition.

In order to characterize the molecular pathways involved in concomitant cellular models, we evaluated AR expression and AR target genes. We observed that only *AR* full-length and *TMPRS2* were the common denominator in the emergence of resistances with any treatment, occurring in both LNCaP and 22RV1 CRPR cellular models. The emergence of resistance was accompanied by constitutive AR activation, detected as an up-regulation of several AR target genes. However, the induction of *AR-V7* and *AR-V9* variants was not necessarily accompanied by greater AR activation. In addition, *AR-V9* up-regulation is always associated to *AR-V7* increase. On the contrary, *AR-V7* induction was not always correlated with higher *AR-V9* expression levels, as Kohli’s study suggested [19]. Interestingly, we also detected that *AR-V9* was associated with resistance to AA in 22RV1 CRPR cellular models, but this correlation was not observed in LNCaP ones. In fact, we observed that resistance to ADT in combination with AA (R-ADT/A) did not involve the up-regulation of *AR-V9*. Importantly, AA was more effective in LNCaP than in 22RV1 cellular models, while Enz was more efficient in 22RV1 cells. In this case, *AR-V9* up-regulation was observed. These results suggest the need to detect the expression levels of both *AR-V7* and *AR-V9* in PCa patients in order to choose the best therapeutic option (AA or Enz).

In conclusion, our work identified that the treatment with AA or Enz could be more effective if used as first-line therapy in androgen-sensitive PCa, since ADT induces additional resistant mechanisms that reduce the efficiency of these drugs as a second-line treatment. Our CRPC cellular models recapitulate the acquisition of cross-resistance between NHAs observed in mCRPC patients. In addition, we suggest the need to identify not only *AR-V7* but also *AR-V9* expression to correctly select the most effective anti-androgen to be administrated.

## Figures and Tables

**Figure 1 cancers-13-01483-f001:**
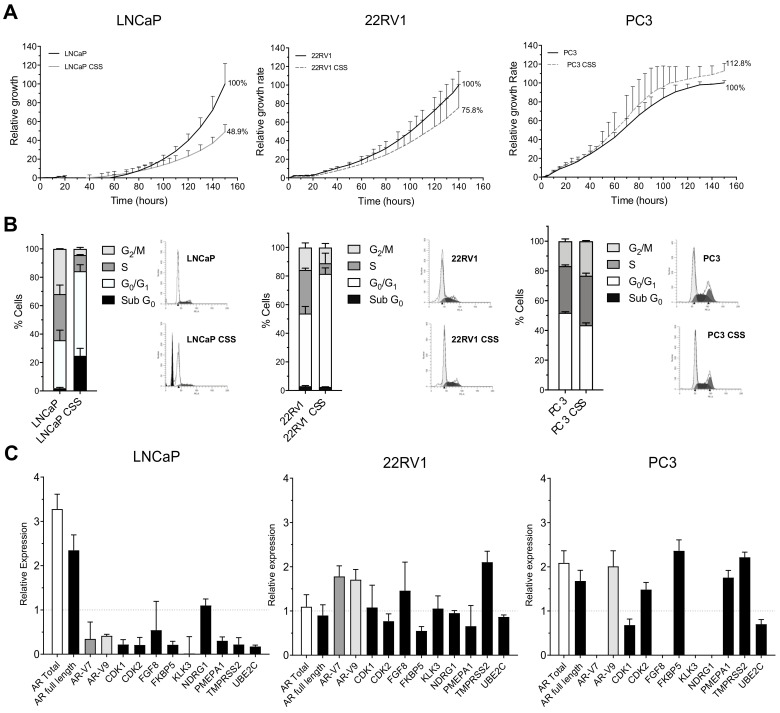
Response to hormonal suppression (ADT) in wild-type PCa cell lines cultured with regular medium or hormone-reduced medium (CSS) for 5 days. (**A**) Analysis of cell proliferation using xCELLigence. Results are normalized considering the final value for control cultures 100%. Data shown correspond to the mean ± SD calculated from triplicates for each condition. (**B**) Cell cycle analysis after 5 days ADT treatment. Bar graphs represent the percentages corresponding to each of the phases of the cell cycle in the different study groups; the error bar corresponds to the SD calculated from the three replicas for each condition. (**C**) Relative expression of androgen receptor (*AR*) isoforms and AR target genes. The results are shown after normalization with respect to endogenous control (GADPH) and referenced to wild-type cell lines grown in regular fetal bovine serum (FBS). The error bars shown correspond to the SD calculated from the replicas.

**Figure 2 cancers-13-01483-f002:**
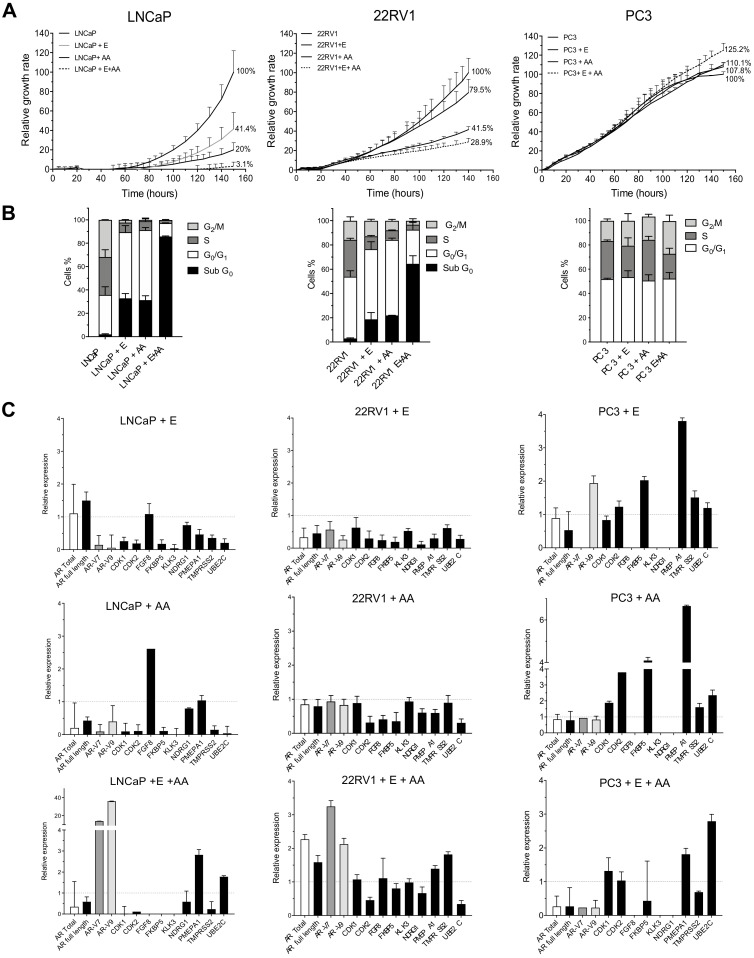
Response to Abiraterone (AA) and Enzalutamide (E) of the wild-type PCa cell lines. Proliferation, cell cycle and quantification of *AR*, *AR-V7*, *AR-V9* and AR target genes. PCa cells were treated for 5 days with 40 μM Enz, 20 μM AA or the combination (40 μM Enz + 20 μM AA). (**A**) Analysis of cell proliferation using xCELLigence. Results are normalized to untreated cells considering their final value after 5-day cultures 100%. Data shown correspond to the mean ± SD calculated from the quadruplicates made for each condition. (**B**) Cell cycle analysis after Enz, AA or the combination (E + AA) treatments for 5 days in LNCaP, 22RV1 and PC-3 cell lines. Stacked bar graphs show the percentages for each cell cycle phase; error bar corresponds to the SD calculated from the triplicates for each experimental condition. (**C**) qPCR analysis for *AR* isoforms and some of AR target genes after Enz, AA or the combination (E + AA) treatments for 5 days in LNCaP (left panels), 22RV1 (middle panels) and PC-3 (right panels). The results are shown after normalization with respect to endogenous control (GADPH) and referenced to the control group (wild-type untreated cells). The error bars correspond to the SD calculated from triplicates.

**Figure 3 cancers-13-01483-f003:**
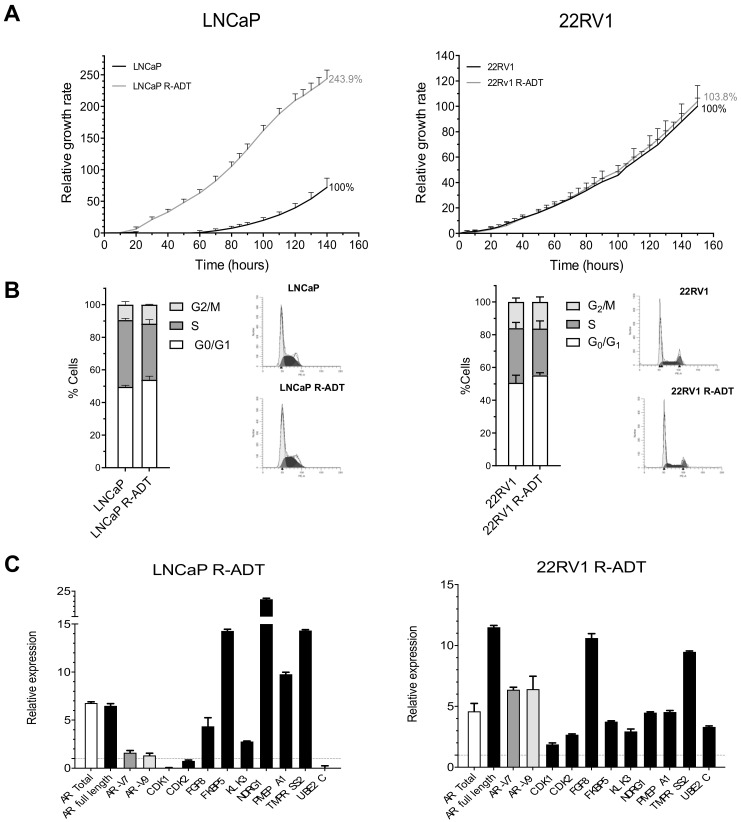
Characterization of ADT-resistant cell lines LNCaP R-ADT and 22RV1 R-ADT (R-ADT model). To obtain ADT-resistant cell lines, sensitive LNCaP and 22RV1 cells were cultured with a hormone-reduced medium (CSS) for 6 months. (**A**) Analysis of cell proliferation using xCELLigence. Results have been standardized considering the final value after 5 days of control cultures to 100%. The error bars shown correspond to the SD calculated from the quadruplicates for each condition. (**B**) Cell cycle analysis in wild-type PCa cell lines grown in regular medium and R-ADT PCa cell lines (LNCaP R-ADT and 22RV1 R-ADT) grown in hormone-reduced medium. Stacked bar graphs show the percentages for each cell cycle phase; error bar corresponds to the SD calculated from triplicates for each experimental condition. (**C**) qPCR analysis for *AR* isoforms and AR target genes after ADT resistance in LNCaP R-ADT (left panel) and 22RV1 R-ADT (right panel) cell lines grown in hormone-reduced medium. The results are shown after normalization with respect to endogenous control (GADPH) and referenced to the wild-type PaC cells. The error bars correspond to the SD calculated from triplicates.

**Figure 4 cancers-13-01483-f004:**
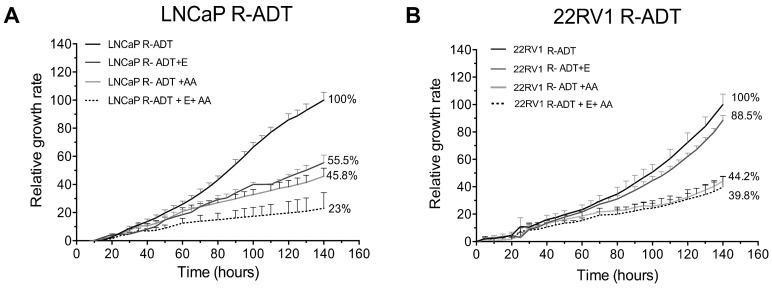
R-ADT cells treated with a NHA as second-line treatment. (**A**) LNCaP R-ADT cells were treated with 40 μM Enz (R-ADT + E); 20 μM AA (R-ADT + AA) and 40 μM Enz + 20 μM AA (R-ADT E + AA) for 5 days. Results have been standardized considering the final value after 5 days of control cultures to 100%. Data shown correspond to the mean ± SD calculated from the quadruplicates made for each condition. (**B**) Results obtained for the 22RV1 R-ADT cell line under the same experimental conditions than LNCaP R-ADT from section A.

**Figure 5 cancers-13-01483-f005:**
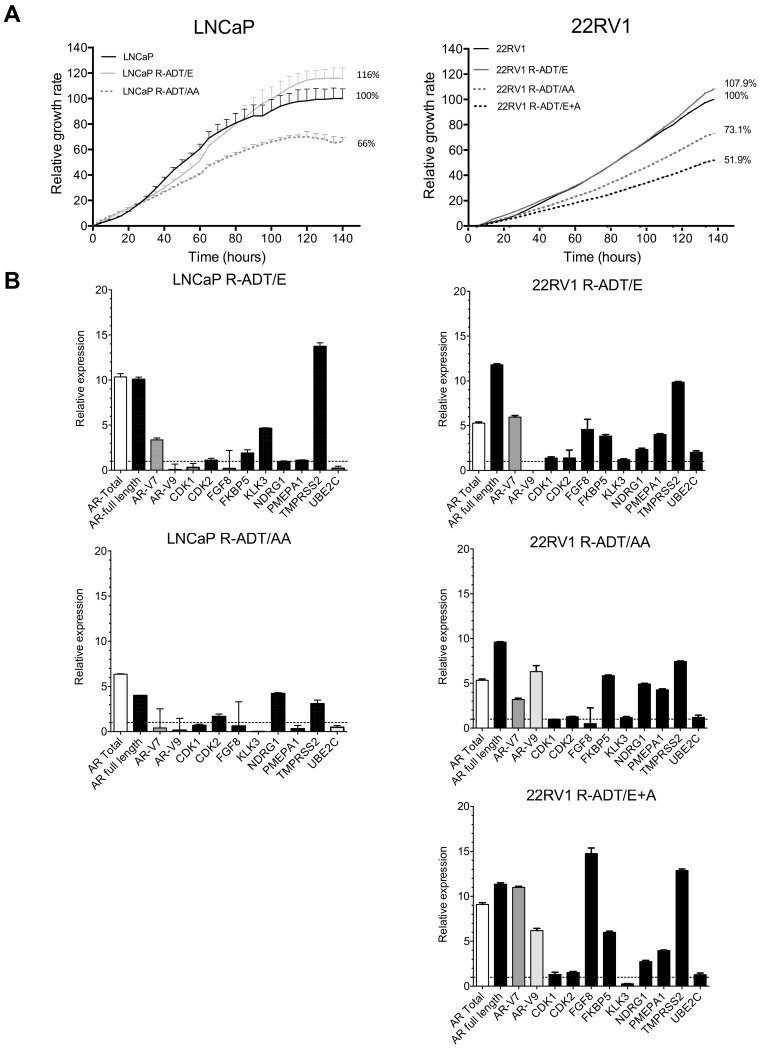
Characterization of NHA-resistant cell lines (R-ADT/E, R-ADT/AA and R-ADT/E + A) by the concomitant use of treatments. LNCaP and 22RV1 wild-type cells were cultured with serum without hormones (CSS) and 40 μM Enz (R-ADT/E); 20 μM of AA (R-ADT/AA) and 40 μM of Enz + 20 μM of AA (R-ADT/E + AA) for 6 months. (**A**) Cell proliferation analysis using xCELLigence in LNCaP (left panel) and 22RV1 (right panel) cell lines. Results have been standardized considering the final value after 5 days of control cultures to 100%. Data shown correspond to the mean ± SD calculated from the quadruplicates made for each condition. (**B**) qPCR analysis for *AR* isoforms and AR target genes after in LNCaP (left panels) and 22RV1 (right panels) concomitant cellular models. The results are shown normalized with respect to endogenous control (GADPH) and referenced to the wild-type cell lines. The error bars shown correspond to the SD calculated from triplicates.

**Figure 6 cancers-13-01483-f006:**
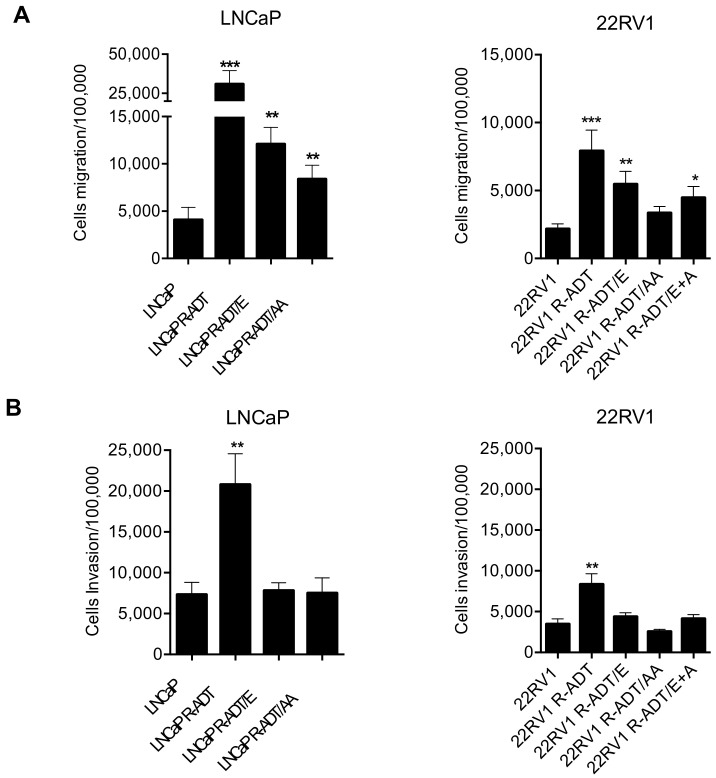
Analysis of migration and invasion assays in the resistant lines derived from LNCaP and 22RV1. (**A**) Migration assays in LNCaP (left panel) and 22RV1 (right panel). (**B**) Invasion assays in LNCaP (left panel) and 22RV1 (right panel). In both sections the histograms represent the number of cells that migrate or invade per 100,000 cells seeded. Data shown correspond to the mean ± SD calculated from the triplicates made for each condition. (* *p* < 0.05; ** *p* < 0.01, *** *p* < 0.001, *t*-Student test). The significant differences shown are with respect to the wild-type cell lines.

**Figure 7 cancers-13-01483-f007:**
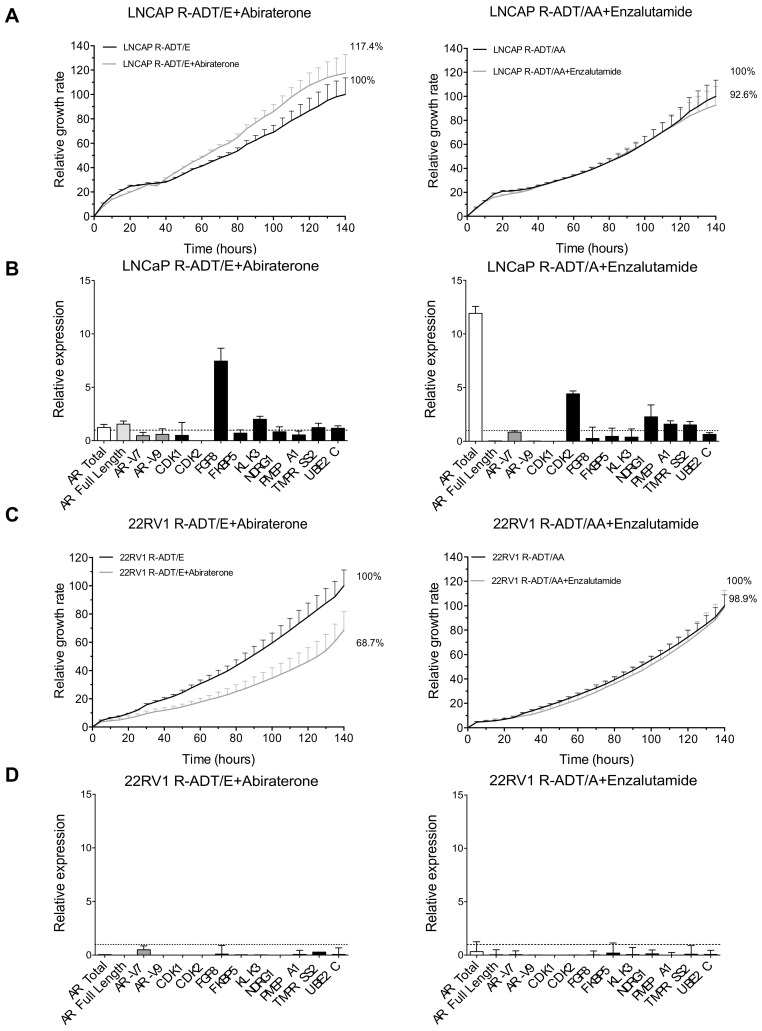
Analysis of cross-resistance between Abiraterone and Enzalutamide from the concomitant cellular models (R-ADT/NHAs + Abiraterone or Enzalutamide). In order to study the cross-resistance to the complementary antiandrogen, R-ADT/E cells were treated with AA (20 μM) (R-ADT/E + Abiraterone), the R-ADT/A cell line was exposed to Enz (40 μM) for 5 days (R-ADT/AA + Enzalutamide), and cell growth was evaluated by xCELLigence. (**A**) Cell proliferation analysis results for LNCaP cell lines using xCELLigence. Results have been standardized considering the final value after 5 days R-ADT/NHAs cell lines to 100%. LNCaP R-ADT/E + Abiraterone (left panel) and LNCaP R-ADT/A + Enzalutamide (right panel). Data shown correspond to the mean ± SD calculated from the quadruplicates made for each condition. (**B**) qPCR analysis for *AR* isoforms and AR target genes. The results are shown normalized with respect to endogenous control (GADPH). The error bars shown correspond to the SD calculated from triplicates. (**C**) Cell proliferation assays and (**D**) qPCR analysis for 22RV1 R-ADT/E + Abiraterone (left panels) and 22RV1 R-ADT/A + Enzalutamide (right panels).

## Data Availability

All the experimental data presented in this article are available in the Results section or the Appendix A in Cancers website. These data are available on request from the corresponding authors. The data are not publicly available due to the privacy policy of our institutions.

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
