# Peer review of "Cross-Resistance to Abiraterone and Enzalutamide in Castration Resistance Prostate Cancer Cellular Models Is Mediated by AR Transcriptional Reactivation"

_cancers, 2021, doi:10.3390/cancers13061483_

Round 1

Reviewer 1 Report

Simon et al. have performed a study of 3 cell models for prostate cancer (PCA3, 22RV1, and LNCaP) and characterized response and resistance to ADT +/-AA +/- Enz.  The paper should be interpreted in the context of current state of knowledge both clinically and with cellular models.  ADT + AA or Enza are proven standard treatments, and the molecular profile and some resistance mechanisms for tumors have been identified (including AR-V7/V9, AR overexpression) through several efforts including SU2C and other sequencing data.  There is also a body of literature characterizing AR amplification/overexpression as a primary resistance mechanism to ADT.  This paper recapitulates a lot of this both pre-clinical and clinical knowledge.  I think the most novel findings are with regard to the possible presence of novel splice variants or AR isoforms, and the analysis of ARv7-v9 interactions. A major limitation is that the cell line models may not be the best for understanding cancer progression biology given their different genetics of each and heterogeneity.

Some other specific comments

  • abstract: "gold" standard
  • abstract and throughout: I would refer to AA and Enza together as novel hormonal agents (NHAs) and not antiandrogens, as AA is a androgen synthesis inhibitor and enzalutamide is an antiandrogen.
  • while there is preclinical evidence (including this paper) of synergy with AA/Enza, this has been rigorously tested in the CRPC setting in an Alliance trial with negative results.

Reviewer 2 Report

The manuscript of Simon et al. describes the generation and characterization of a new  experimental CRPC model, based on the androgen sensitive/responsive PCa cell lines LNCaP and 22Rv1. The paper is well written, data are clearly presented. I suggest to accept the manuscript after minor revisons.

MINOR POINTS:

  • Intoduction: page 2, paragraph 3: <In addition, AR-V9 shares cryptic exon 3 (CE3) with AR-V7 ….. > is missleading and should be replaced by <In addition, AR-V9 was shown to share a common 3‘  terminal cryptic exon with AR-V7 …..>

  • Introduction: page 2, last paragraph: replace the term <androgen-dependent> by <androgen sensitive>.

  • The relative growth rates of resistant cell lines (R-ADT/E, R-ADT/AA, Fig. 5A) were tested at enzalutamide/abiraterone concentrations of 40 and 20 µM, respectively. Are there also dose response curves available?

  • As shown in J Cell Biochem 2019; 120: 16711 enzalutamide concentrations >30 µM are able to induce antiproliferative off-target effects in PC-3 cells. The authors should discuss this finding in the manuscript.

  • Expression levels of AR and/or AR-V7 in LNCaP, 22Rv1, LNCaP R-ADT and 22Rv1 R-ADT determined by qRT-PCR must be complemented by Western Blots, as mRNA levels do not always reflect protein levels.

  • LNCaP cells carry a progesterone responsive AR-mutant (T878A) that induces an abiraterone resistance (Clin Cancer Res 2015; 21:1273). Please discuss/elaborate the implications of this finding for the current experimental model.

Reviewer 3 Report

I find this a solid and interesting study of the achievement of CRPC in vitro, but I have a few comments and suggestions to improvements as follows:

  1. M&M section 2.2: The cell medium consists of RPMI, but it is not stated whether it contains phenol red or not. It is well known that many cell lines can use this for synthesis of steroids. Media containing phenol red is often sold also in a version without and if this is used it should be mentioned in this section. If not, it would be more accurate to describe the media used in this study, as hormone reduced, not hormone deficient.
  2. Is it one culture of each strain that has been selected using the different conditions or have the experiments been repeated? The different treatments seem to affect the cell lines differently and I am curious if this is due to the different genetic backgrounds of each, or if it could be due to accidental mutations that have been selected over time. I also lack a discussion on why the different cell lines seems to react so differently to the different treatments.
  3. Section 2.8: I am not familiar with the term “Retrotranscription” – should it be reversed transcription instead?
  4. Section 2.10: Should “paired Student’s test” be “paired Student’s t-test”?
  5. Fig 1A: The markings used for the wt and the CSS cell lines are quite similar and a bit hard to distinguish both when I look at the figures in my computer and when I print it out – could this be improved to facilitate for the reader? The font size is also quite small.
  6. Fig 1A PC3: The only comment I found about this figure is that no reduction of growth rate was detected confirming ADT treatment resistance. However, what is the explanation for the growth rate increasing (more than 10% as far as I can see) when reducing the concentration of hormones in the media? Maybe this is also in line with the results presented in Fig 2A, where all treatments actually increases the growth rate in contrast to the results from the other cell lines where you see a pronounced reduction in growth rate.
  7. Fig 1B: I really appreciate that the authors have included the PI histogram plots, but unfortunately, I find the size and quality of them a bit deficient. When I try to increase the size on my computer, they get quite blurry. I also think there is some kind of colour coding of them that I find a bit confusing. If it were impossible to increase the size in the present figure, maybe it would be an idea to make a separate figure with them and include as a supplementary figure. The problem with the colour coding is also valid for the columns, where it seems to be more stringent in the following figures.
  8. Fig 3: It would facilitate for the reader if the authors could include a small note how the cells are cultured in the figure legend – I fell quite unsure if the cells in the experiment are grown in ordinary media or using CCS-media. Do the LNCaP R-ADT cells have such increased growth rate during ADT conditions or under all conditions? In addition, why is LNCaP in Fig 3B differ from Fig 1B?
  9. Fig 5A: I cannot see any marking for LNCaP R-ADT/E in the figure.
  10. The authors end the discussion with the conclusion that it is necessary to measure not only AR-V7 but also AR-V9 expression in order to select the most effective anti-androgen – I would like to hear more about the reason behind this as I can only find one example where AR-V9 is increased when not AR-V7 is concomitantly increased and that is in Fig 2C PC3+E. But the AR dependent genes showing increased expression there shows even higher expression when in PC3 + AA, although no increase in AR-V9, or any of the other AR variants, is detected.
  11. How stable are the drugs used in the study?
    How often did you have to replenish the media during the incubations?

Reviewer 4 Report

General comments:

This paper further elucidates the mechanisms of resistance with enzalutamide or abiraterone

Specific comments:

  1. Great work on establishing overexpression of AR full-length and AR target genes, but not necessarily AR-V7 and/or AR-V9 isoforms as one of the potential etiology of mechanisms of resistance. Was there an attempt to further elucidate other mechanisms of resistance other than AR pathways?
  2. Perhaps more clinically relevant to explore pathways of resistance with the newer/novel antiandrogens apalutamide or darolutamide, since enzalutamide resistance mechanisms have already been well described

Round 2

Reviewer 1 Report

Authors have addressed the raised issues from review.

Author Response

We thank the reviewer for his/her academic review that substantially improved our current version of the manuscript. We are very happy that we have been able to raise all his/her previous comments and suggestions.

Reviewer 4 Report

It appears as though the limitations in methodology is set since this is part of the first author's thesis as described as well in the rebuttal.

Author Response

We thank the reviewer for his/her academic review and the kinds words that we received from the beginning of the revision process. We are pleased that the reviewer has understood our explanations to his/her initial comments replied in our previous rebuttal letter.

We agree that the inclusion of newer antiandrogens, such as apalutamide or darolutamide, would be a very interesting approach but it is out of the scope of this manuscript. We would like to follow this interesting suggestion in the next future in our laboratories.